# SWE-bench Goes Live!

**Linghao Zhang**[1][*]   **Shilin He**[1][†]   **Chaoyun Zhang**[1]   **Yu Kang**[1]   **Bowen Li**[2]
**Chengxing Xie**[2]   **Junhao Wang**[1]   **Maoquan Wang**[1]   **Yufan Huang**[1]   **Shengyu Fu**[1]
**Elsie Nallipogu**[1]   **Qingwei Lin**[1]   **Yingnong Dang**[1]   **Saravan Rajmohan**[1]   **Dongmei Zhang**[1]
[1]Microsoft    [2]Shanghai Artificial Intelligence Laboratory

## Abstract

The issue-resolving task, where a model generates patches to fix real-world bugs, has emerged as a key benchmark for evaluating the capabilities of large language models (LLMs). While SWE-bench has become the dominant benchmark in this domain, it suffers from several limitations: it has not been updated since its release, is restricted to only 12 repositories, and relies heavily on manual effort for constructing test instances and setting up executable environments, significantly limiting its scalability. We present **SWE-bench-Live**[3], a *live-updatable* benchmark designed to address these limitations. SWE-bench-Live currently includes 1,890 tasks derived from real GitHub issues created since 2024, spanning 223 repositories. Each task is accompanied by a dedicated Docker image to ensure reproducible execution. Additionally, we introduce an automated curation pipeline that streamlines the entire process from instance creation to environment setup, removing manual bottlenecks and enabling scalability and continuous updates. We evaluate a range of state-of-the-art models and agent frameworks on SWE-bench-Live, offering detailed empirical insights into their real-world bug-fixing capabilities. By providing a fresh, diverse, and executable benchmark grounded in live repository activity, SWE-bench-Live supports reliable, large-scale assessment of code LLMs and code agents in realistic development settings.

## 1   Introduction

Large language models (LLMs) have fundamentally reshaped the landscape of software engineering [7], powering tools such as Cursor [4] and GitHub Copilot [5] that are now integral to modern development workflows. These models have transformed key stages of the software development lifecycle—automated code generation, bug detection, and issue resolution—leading to substantial gains in developer productivity. To systematically assess LLM capabilities across these tasks, a variety of curated benchmarks have been developed, including HumanEval [3], MBPP [2], SWE-bench [10], DI-Bench [32], and OpenRCA [19]. These benchmarks are instrumental in identifying both the strengths and limitations of LLMs in diverse programming and maintenance settings.

Among them, SWE-bench [10] and its variants, such as Multimodal SWE-bench [23] and Multi-SWE-bench [26], have become standard for evaluating LLMs on the issue resolution task, where models are required to comprehend complex codebases, interact with execution environments, and generate patches that fix real-world issues. However, as LLMs evolve rapidly, existing benchmarks exhibit several critical limitations that undermine their continued utility:

---

[*]Work done during the internship at Microsoft

[†]Shilin He is the corresponding author

[3]Homepage: `https://swe-bench-live.github.io/`, Code: `https://github.com/microsoft/SWE-bench-Live`, and Dataset: `https://huggingface.co/SWE-bench-Live`

39th Conference on Neural Information Processing Systems (NeurIPS 2025) Track on Datasets and Benchmarks.

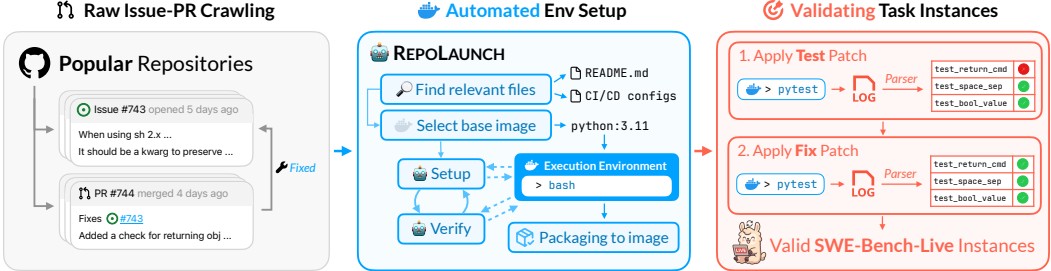

Figure 1: The automatic construction pipeline of SWE-bench-Live.

Table 1: Comparison with existing issue resolving benchmarks.

| Dataset | Date | #Instances | #Repository | Real/Synthetic | Curation |
|---|---|---|---|---|---|
| SWE-bench [10] | Oct, 2023 | 2,294 | 12 | Real | Manual |
| SWE-bench-Verified [13] | Aug, 2024 | 500 | 12 | Real | Manual |
| SWE-Gym [14] | Dec, 2024 | 2,438 | 11 | Real | Manual |
| Multi-SWE-bench [26] | Apr, 2025 | 1,632 | 39 | Real | Manual |
| SWE-smith [24] | Apr, 2025 | 50,000 | 128 | Synthetic | Semi-manual |
| **SWE-bench-Live** (Ours) | April, 2025 | **1,890 (since 2024)** | **223** | **Real** | **Automatic** |

1. **Staleness.** SWE-bench and its derivatives have not been updated since their initial releases, making them static benchmarks. Because LLMs are trained on massive inscrutable corpora, these static datasets are at risk of data contamination, as they could have be been unpurposely included in model training data. This raises concerns about whether newer models are making truly generalizable progress or merely memorizing benchmark content, reducing the benchmarks' effectiveness in distinguishing model capabilities.

2. **Limited repository coverage.** These benchmarks draw from a small set of repositories, limiting diversity in codebases, domains, and programming practices (see Table 1 for details). This narrow scope weakens the generalizability and robustness of evaluations.

3. **Heavy reliance on manual effort.** Constructing instances for SWE-bench-like task istances involves substantial human labor: identifying appropriate issue-resolution pairs, locating relevant tests, configuring runnable environments, composing test commands, and validating the full workflow.[4] This process is resource-intensive and creates scalability bottlenecks.

To address these challenges, we introduce SWE-bench-Live, a live and scalable benchmark built for evaluating LLMs on real-world issue resolution tasks. In contrast to recent efforts such as LiveCodeBench [9], which target algorithmic programming problems, SWE-bench-Live is the first live-updating benchmark designed for complex, repository-level tasks that demand multi-file reasoning, environment setup, and reproducible execution. Figure 1 illustrates the construction pipeline of SWE-bench-Live. At the core of our framework is REPOLAUNCH, a fully automated pipeline that eliminates manual bottlenecks by streamlining the entire process—from issue mining to environment packaging. More specifically, REPOLAUNCH leverages an agentic and end-to-end workflow to setup the Docker environment by identifying relevant instruction files, selecting base images, installing necessary dependencies, building the project, and validating its test suite. This automation enables continuous updates, broad repository coverage, and large-scale dataset expansion. To date, REPOLAUNCH has been capable of supporting all major programming languages (including C, C++, C#, Python, Java, JS/TS, and Go) and is able to build on both Linux and Windows platforms, making it a versatile environment creation agent.

The initial release of SWE-bench included 1,319 issue-resolution tasks sourced from 93 repositories. Through continuous monthly updates, our current release contains 1,890 issue-resolution tasks sourced from real-world GitHub issues created since 2024, spanning 223 repositories. Compared to existing benchmarks, this represents a significant leap in freshness, diversity, and scale (see Table 1).

We evaluate three leading agent frameworks (i.e., OpenHands [16], SWE-Agent [21], and Agent-less [17]) in combination with four state-of-the-art LLMs (namely, GPT-4.1, GPT-4o, Claude 3.7

---

[4]For instance, it take about one year for Multi-SWE-bench [26] to create 1,632 benchmark instances with 68 expert annotators.

Sonnet, and DeepSeek V3). Consistent with performance rankings reported on SWE-bench Verified,[5] we observe that OpenHands, when paired with Claude 3.7 Sonnet, achieves the highest performance on SWE-bench-Live. However, its overall scores are significantly lower compared to those achieved on SWE-bench Verified. To explore this discrepancy further, we conduct a controlled comparison and find that the same agent-LLM pair consistently performs worse on SWE-bench-Live than on SWE-bench. This finding suggests that existing models may be overfitting to static benchmarks like SWE-bench, underscoring the importance of developing more dynamic and diverse evaluation settings, such as those provided by SWE-bench-Live.

Our main contributions are summarized as follows:

- We introduce SWE-bench-Live, a contamination-resistant, reproducible, and continuously updatable benchmark tailored to real-world issue resolution tasks. It reflects the dynamic nature of software development and offers broader repository coverage compared to prior benchmarks.

- We propose REPOLAUNCH, a fully automated pipeline for benchmark construction that seamlessly integrates data curation, environment setup, and test validation into a cohesive and scalable system.

- Through experimental evaluation, we observe the suboptimal performance of leading agent frameworks on SWE-bench-Live, highlighting significant opportunities for improvement on the contamination-free benchmark.

## 2  Related Work

**Coding Benchmarks.**    Early benchmarks for program synthesis and bug fixing focused on *single-file, synthetic* tasks such as HumanEval [3] and MBPP [2], which do not reflect the complexity of real repositories. To move closer to practice, SWE-bench [10] introduced the *issue-resolving* task, requiring a model to generate a validated patch for a GitHub repositories issue. Numerous extensions have since appeared—including Multimodal SWE-bench for JavaScript and UI screenshots [23], Multi-SWE-bench for multiple languages such as Java and Rust [26]. Despite their impact, all of these datasets are *static*: they are collected once, cover at most a few dozen repositories, and depend on labor-intensive environment construction. These yield two limitations. First, models can overfit to the fixed test set, inflating apparent progress. Second, public tasks may lead to *data contamination*, where benchmark instances leak into pre-training corpora [31, 8]. Recent "live" datasets such as LiveCodeBench [9] mitigate contamination by streaming *algorithmic* problems after their release dates, yet they do not address the harder *repository-level* setting that demands multi-file reasoning and execution inside a faithful environment. SWE-bench-Live is the first open, continuously updating benchmark that fulfills these requirements.

**Coding Agents.**    On top of the above benchmarks, a recent line of work has been working creating autonomous *code agents* that search, edit, and test large codebases. Representative systems include SWE-Agent [22], OpenHands [16], Agentless [17], and training frameworks that synthesize thousands of SWE-bench-like instances [15, 24, 18]. These agents report remarkable headline numbers, yet their evaluations rely almost exclusively on static offline datasets. As a consequence, improvements may partially stem from memorisation of leaked solutions or configuration quirks, rather than genuine advances. SWE-bench-Live closes this gap by pushing agents to fix *previously unseen, continuously arriving* real-world bugs under fully reproducible Docker images, it reveals failure modes hidden by stale test suites and provides a trustworthy yard-stick for code agents and LLMs.

## 3  SWE-bench-Live

Targeting the issue resolution task on real-world GitHub repositories, SWE-bench serves as a practical proxy for evaluating the coding capabilities of LLM-based systems. The issue resolving task is defined as follows: given a code repository and an associated issue, an approach (e.g., LLM agent) is required to generate a patch that resolves the issue and passes the test cases (see Appendix B for details).

---

[5]https://www.swebench.com/

While SWE-bench-Live adopts the same task definition as SWE-bench, it introduces a *novel, fully automated pipeline* that enables scalable and continuously updatable benchmark construction. This automation allows for a larger number of up-to-date instances and broader repository coverage.

**Pipeline Overview.**   As shown in Figure 1, the construction of SWE-bench-Live follows a three-stage pipeline. First, starting from popular repositories, we identify GitHub issues that are resolved by a pull request (PR). Next, we apply the proposed REPOLAUNCH—an agentic approach that automatically sets up an container-based execution environment for each candidate instance. Finally, we perform multiple rounds of test execution for each instance to validate whether it consistently exhibits the expected issue-resolving testing behavior, and finalize the valid instances.

Thanks to its fully automated pipeline, SWE-bench-Live can be maintained with minimal–ideally zero–manual effort. We plan to update SWE-bench-Live on a monthly basis, continually providing the community with an up-to-date evaluation dataset. This enables contamination-free, rigorous assessment of AI systems' issue-resolving capabilities in a constantly evolving real-world setting.

### 3.1   Raw Issue–PR Crawling

The first phase of the SWE-bench-Live pipeline involves collecting real-world issue–pull request (PR) pairs from popular open-source GitHub repositories.

**Repository Selection.**   We focus on Python repositories for the initial release of SWE-bench-Live, aligning with SWE-bench and other prior benchmarks due to its popularity. The selection process includes three filtering stages: *(i)* We first queried GitHub API for repositories with over 1,000 stars and Python set as the primary language. This initial query yielded 8,577 repositories as of April 2025. *(ii)* We then refined this set by requiring each repository to have more than 200 issues and pull requests, over 200 forks, and at least 60% of its codebase written in Python. This reduced the pool to 3,316 repositories. *(iii)* Finally, to comply with licensing requirements, we retained only repositories containing a valid open-source license, resulting in a final selection of 2,609 repositories.

**Issue–PR Pair Extraction.**   From the selected repositories, we adopt the collection script from SWE-bench to extract issue and its associated PR. Meanwhile, the pull request must modify the repository's test suite–i.e., a "test patch", which will serve as the evaluation targets. We also incorporate improvements from SWE-Fixer [18], which introduces more robust heuristics to improve the effectiveness of issue–PR pair identification and reduce reliance on the brittle string-matching method. To reduce the risk of data leakage, SWE-bench-Live prioritizes recency by including only issues created after January 2024 in our initial release.

### 3.2   REPOLAUNCH: Automated Execution Environment Setup

The "raw" issue–PR pairs remain at the textual and plain code level. To support subsequent test-based evaluation, it is required to provide an *execution environment* capable of running tests locally and producing execution feedback. In the context of issue-resolving benchmarks, the execution environment is critical for test-based evaluation.

However, preparing such execution environments is widely recognized as the **most labour-intensive step** in constructing issue-resolving datasets. In prior work, including SWE-bench [10] and SWE-Gym [14], environment setup has been performed entirely by humans. For example, SWE-Gym reports that building execution environments required over 200 hours of manual effort, underscoring a significant scalability bottleneck. Notably, even repository-level environments are insufficient: different commits within the same repository may depend on different libraries or configurations, necessitating environment construction at the *snapshot* level. SWE-bench partially mitigates this by building environments per version tag, but the granularity remains coarse and relies on manual labor.

To address this bottleneck, we introduce an agent-based framework REPOLAUNCH, which automatically creates a fully functional execution environment for each issue instance. For any given *repository snapshot*, REPOLAUNCH produces a Docker container that installs all required dependencies, builds the project, and validates its test suite. This containerized instance serves as the foundation for running and evaluating model-generated patches.

**Repository Snapshots and Environment Definition.** A repository snapshot corresponds to the codebase at the base commit associated with an issue. The goal is to recreate an environment faithful to that moment in time. We define a valid execution environment as a Docker container where *(i)* the codebase is correctly installed from source, and *(ii)* the repository's test suite passes with zero or tolerable failures. This environment is essential for test-based evaluation, providing the ground truth mechanism to verify whether the issue has been resolved.

REPOLAUNCH follows an LLM-driven, agentic workflow [28, 27] inspired by how human developers set up unfamiliar projects, as shown in Figure 1. The process proceeds in five steps:

- **Relevant Files Identification.** The first step is to identify relevant files in the repository–such as CI/CD pipelines and README files that are likely to contain useful information for setting up the environment (a detailed list is provided in the Appendix H).

- **Base Image Selection.** Given the full content of the relevant files, this step is to select a suitable base Docker image based on the information provided in the repository. This involves correctly identifying the programming language and SDK version used in the repository (e.g., `python:3.11`). A container is instantiated from the chosen image, and a persistent bash session is launched.

- **Interactive Environment Setup.** The setup process is carried out by an agent whose goal is to successfully execute and pass all test cases in the repository's test suite within the container. The agent interacts with the bash session by issuing commands and receiving feedback such as exit codes and outputs. It follows the ReAct design [25], iterating over *Thought → Action → Observation* [30, 29], mimicking a developer's reasoning and trial process. The agent can also search the web or query the issue tracker for troubleshooting.

- **Verification.** Once the setup agent determines that the environment has reached a satisfactory state or a step limit is reached, control is transferred to a verifying agent. The agent attempts to generate the appropriate test command and execute it. The execution results are evaluated with the agent to check if all test cases passed. If test failures occur, the results are fed back to the setup agent for further refinement. If all tests pass, the environment is considered valid.

- **Finalization.** Upon successful validation, the container is committed as a Docker image, producing a instance-level execution environment for reuse.

**Challenges of Version Incompatibility.** A major challenge when setting up out-of-date repositories is the "dependency version drift" issue. When dependencies are not pinned to specific versions, tools like `pip` by default will resolve to the latest package versions, which often introduce backward-incompatible issues and make the environment setup fail. To address this, we implement an *time-machine* mechanism by forcing the package installation tool to only look at valid versions released no later than the current base commit timestamp. Specifically, we modified the `pip` default index server to a proxy which fetches those valid package versions. This simple but effective strategy prevents the "future" version incompatibilities and significantly improves setup success rates.

We open-source REPOLAUNCH to benefit the community. While designed for automated benchmark construction, REPOLAUNCH can also assist developers in quickly setting up environments for unfamiliar codebases. Its ability to replicate historical setups and automatically resolve environment dependencies positions it as a practical tool with broader applicability beyond benchmarking.

To date, REPOLAUNCH has been capable of supporting all major programming languages (including C, C++, C#, Python, Java, JS/TS, and Go) and is able to build on both Linux and Windows platforms, making it a versatile environment creation agent.

### 3.3 Validating Task Instances

To ensure the quality of the benchmark, each task instance is validated to confirm that the associated PR effectively resolves the issue it is intended to fix. The validation is based on analyzing changes in the test suite results before and after applying the PR's patch. Specifically, we focuses on identifying two key behaviors in the test outcomes:

- `FAIL_TO_PASS` transitions: Tests that were initially failing (`FAILED` or `ERROR`) and later passing (`PASSED`) after the patch is applied. These yield that the patch addresses the issue effectively.

- `PASS_TO_PASS` transitions: Tests that were both passing before and after the patch is applied. These transitions demonstrate that the patch does not break unrelated functionality.

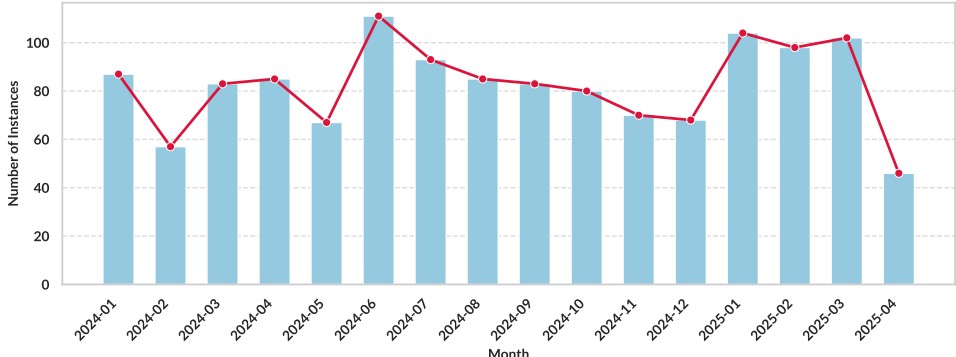

Figure 2: Temporal distribution of issue creation times in SWE-bench-Live.

To identify these transitions, the test results (as logs) are collected both before and after applying the PR's patch. By comparing individual test outcomes between the two runs, we determine how the patch affected specific tests. We designed framework-specific (e.g., tox, pytest) parsers to interpret test outputs reliably, as different testing tools may produce logs in various formats. For a task instance to be included in the benchmark, it must exhibit at least one `FAIL_TO_PASS` transition. Instances lacking such a transition are excluded because they do not demonstrate effective bug resolution. Additionally, to ensure reproducibility and avoid issues caused by test flakiness, the validation process is repeated multiple times. Only instances with consistent results across all runs are retained. This approach ensures that all task instances are grounded in evidence of real-world bug fixes and preserves stable behaviors, resulting in a robust benchmark for evaluating automated bug-fixing solutions.

### 3.4 SWE-bench-Live Statistics

The initial release of the SWE-bench-Live dataset consists of 1,319 task instances collected from real-world issues and pull requests across 93 open-source Python repositories. To ensure freshness and reduce the risk of data contamination from pretraining, we restrict the dataset to issues created between January 1, 2024, and April 20, 2025. As shown in Figure 2, the temporal distribution is generally uniform, indicating consistent coverage of issues over time. We plan to update the dataset on a monthly basis to reflect the evolving software landscape and continuously provide new instances.

Table 2 summarizes key statistics at both the repository and instance levels. At the repository level, projects vary in size, with an average of 85k lines of Python code and 423 files. At the instance level, we report metrics of the gold patches—including the number of edited files, hunks, and lines—as heuristic indicators of task complexity. These statistics suggest that SWE-bench-Live tasks reflect realistic, non-trivial bug fixes that challenge code understanding, reasoning, and manipulation capabilities of LLMs. Additionally, we record the number of test cases that transition from failure to pass (F2P) and those that consistently pass (P2P), which form the basis of test-based evaluation.

**Repository Diversity.** To ensure broad applicability, SWE-bench-Live includes repositories from diverse application domains. As shown in Figure 3, we manually categorized each repository based on its primary functionality—such as AI/ML, DevOps, Web development, and others. This diversity helps evaluate LLMs across varied software stacks and bug types, enhancing the benchmark's representativeness of real-world usage scenarios.

**Lite Subset.** To support lightweight experimentation, we construct a lite subset of SWE-bench-Live by sampling 50 instances per month from issues created between October 2024 and March 2025. This results in a compact set of 300 instances that balances recency, diversity, and evaluation efficiency.

**Comparison with Existing Benchmarks.** Table 1 compares SWE-bench-Live with several existing issue-resolution benchmarks. Unlike SWE-bench and its variants, which require extensive manual curation and cover a limited set of repositories, SWE-bench-Live is the first to offer an **automatically** constructed, continuously updatable benchmark. It covers a broader set of repositories (93 in total), while preserving the use of real issues and test-based evaluation. Compared to synthetic datasets like SWE-smith, which may not fully capture the complexity of human-written code and bugs,

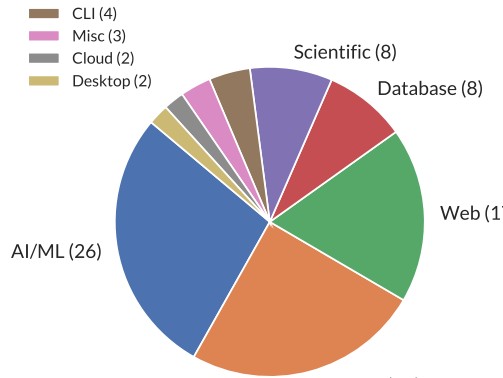

Figure 3: Repository classifications.

Table 2: Statistics of SWE-bench-Live

| Level | #Item | Average | Median |
|---|---|---|---|
| Repo | Repositories | 93 | |
| | LoC[*] | 85k | 52k |
| | Files[*] | 423 | 222 |
| Instance | Instances | 1319 | |
| | Files[†] | 3.3 | 2 |
| | Hunks[†] | 9.0 | 3 |
| | Lines[†] | 102.6 | 24 |
| | F2P test cases | 5.4 | 1 |
| | P2P test cases | 2953.4 | 1865 |

[*]*Only count Python code.* [†]*Stats of gold patch.*

SWE-bench-Live maintains fidelity to real-world development workflows. Its unique combination of automation, realism, and diversity fills a critical gap of the LLM evaluation for software engineering.

## 4 Experiments

### 4.1 Setups

**Agents and Model Selection.** To evaluate the effectiveness of our proposed SWE-bench-Live, we conduct experiments using three representative agent frameworks. These include the general-purpose coding agent **OpenHands** [16] (paired with CodeAct), as well as two agents specifically designed for issue-resolving tasks: **SWE-Agent** [21] and **Agentless** [17]. For OpenHands, we set a maximum of 60 iterations per instance. For SWE-Agent, we limit the number of LLM calls to 100 per instance to maintain computational efficiency. For Agentless, we largely follow the original pipeline, which consists of two main stages: issue localization and patch generation. However, we *omit* the reranking stage based on regression testing, as supporting this step on SWE-bench-Live would require substantial infrastructure adaptation and is beyond the scope of this study. Consequently, both the localization and repair stages in our Agentless evaluation produce a single sample without reranking and patch selection. We refer to this simplified evaluation protocol as **s-Agentless** throughout the paper. We test these agents using four recent state-of-the-art LLMs, covering both proprietary and open-source models: GPT-4o [11] (gpt-4o-2024-11-20), GPT-4.1 [12] (gpt-4.1-2025-04-14), Claude 3.7 Sonnet [1] (claude-3-7-sonnet-20250219), DeepSeek V3 [6] (DeepSeek-V3-0324), and Qwen3-Coder-480B-A35B[6] [20].

**Evaluation Metrics.** Following the evaluation protocol of SWE-bench [10], we adopt the **Resolved Rate (%)** as our primary metric. This measures the proportion of issues successfully resolved by the agent across all task instances. We also report the **Patch Apply Rate (%)**, which indicates the percentage of generated patches that are syntactically correct and can be successfully applied to the codebase without errors. Additionally, we measure the **Localization Success Rate (%)** at the file level. This reflects whether the set of files modified by the generated patch matches the gold patch.

### 4.2 Performance on SWE-bench-Live

We report the performance of all agent–model combinations on the Lite subset of SWE-bench-Live in Table 3. Meanwhile, Table 4 presents the results of the top three combinations selected based on Lite performance, evaluated on the full version of SWE-bench-Live.

We observe that the same methods achieve substantially higher scores on SWE-bench compared to their performance on SWE-bench-Live, despite both benchmarks targeting the same issue-resolving task with identical settings. For example, recent state-of-the-art agents and models report a resolved rate exceeding 70% on the SWE-bench Verified subset[7]. In contrast, the highest resolved rate on

---

[6]Thanks to the Qwen team for evaluating and reporting the results.
[7]https://www.swebench.com/

Table 3: Performance on SWE-bench-Live (Lite subset).

| Models | Resolved (%) | Apply (%) | Loc. Suc. (%) |
|---|---|---|---|
| **OpenHands** | | | |
| GPT-4o | 7.00 | 72.00 | 30.33 |
| GPT-4.1 | 11.33 | 59.33 | 28.67 |
| Claude 3.7 Sonnet | 17.67 | 84.00 | 48.00 |
| DeepSeek V3 | 13.00 | 81.00 | 38.33 |
| Qwen3-Coder-480B-A35B | 24.67 | 97.00 | 30.00 |
| **SWE-agent** | | | |
| GPT-4o | 10.00 | 93.33 | 40.33 |
| GPT-4.1 | 16.33 | 95.00 | 47.33 |
| Claude 3.7 Sonnet | 17.67 | 84.67 | 46.33 |
| DeepSeek V3 | 15.33 | 92.00 | 44.00 |
| **s-Agentless** | | | |
| GPT-4o | 11.67 | 91.67 | 37.67 |
| GPT-4.1 | 12.00 | 84.33 | 39.00 |
| Claude 3.7 Sonnet | 11.33 | 68.00 | 30.00 |
| DeepSeek V3 | 13.33 | 83.33 | 40.67 |

Table 4: Performance of top-3 performing Agent + Model combinations on SWE-bench-Live.

| Agent / Model | Subset | Resolved (%) | Apply (%) | Loc. Suc. (%) |
|---|---|---|---|---|
| OpenHands / Claude 3.7 Sonnet | Lite | 17.67 | 84.00 | 48.00 |
| | Full | 19.25 | 85.89 | 48.29 |
| SWE-agent / GPT-4.1 | Lite | 16.33 | 95.00 | 47.33 |
| | Full | 18.57 | 94.54 | 49.50 |
| SWE-agent / Claude 3.7 Sonnet | Lite | 17.67 | 84.67 | 46.33 |
| | Full | 17.13 | 89.15 | 45.86 |

SWE-bench-Live is only 24.67%. Considering that the experimental setups on the SWE-bench leaderboard often involve dramatically high rollout numbers or iteration efforts, we specifically re-ran the best performing combination, OpenHands with Claude 3.7 Sonnet, on the SWE-bench verified subset using the exact same setups as in our experiments. The resulting resolved rate reached 43.20%, more than twice the score achieved on SWE-bench-Live. This is a particularly interesting phenomenon, as it highlights the challenges of constructing a benchmark that can objectively measure an AI system's ability to resolve arbitrary and previously unseen issues. It also raises concerns about potential overfitting to SWE-bench. Similar phenomena are also observed in other existing issue-resolving datasets: the best-performing method in Multi-SWE-bench achieves a resolved rate of only 19.32%, while the highest score reported in OmniGIRL is as low as 8.6%.

To investigate this, we further categorize the instances in SWE-bench-Live based on their repository origin. Specifically, 216 instances are derived from 8 repositories that were originally included in SWE-bench, which we refer to as *From SWE-bench Repos*. The remaining 1,103 instances are sourced from repositories not previously used in SWE-bench and are denoted as *From Non-SWE-bench Repos*. As shown in Table 5, although the Non-SWE-bench repositories are generally simpler with fewer files and lower code volume, the best-performing agent–model pair achieves a higher resolved rate of 22.96% on SWE-bench Instances, compared to only 18.89% on the Non-SWE-bench ones. This reinforces the hypothesis that existing agents may be overfit or implicitly optimized for the SWE-bench repositories, further motivating the need for continuously updated, contamination-resistant benchmarks like SWE-bench-Live.

Table 5: SWE-bench vs. Non-SWE-bench.

| Instances | Avg. repo files | Avg. repo loc | Resolved (%) |
|---|---|---|---|
| From SWE-bench Repos | 744 | 223k | 22.96 |
| From Non-SWE-bench Repos | 383 | 68k | 18.89 |

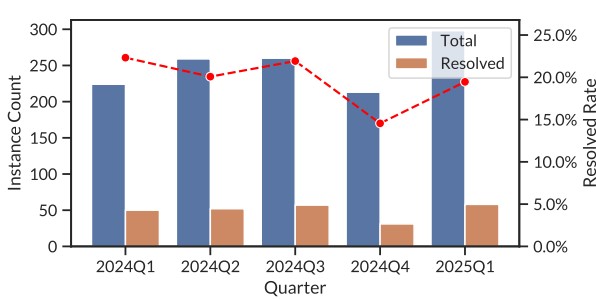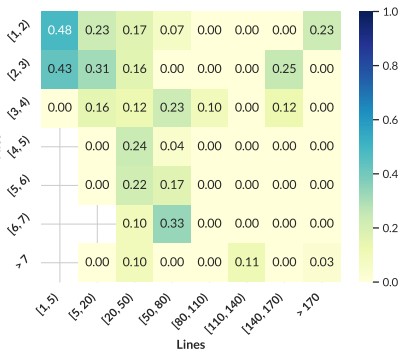

Figure 4: Resolved rate in relation to the creation date of instances. (OpenHands / Claude 3.7 Sonnet on full set)

Figure 5: Resolved rate in relation to the difficulty of instances. (OpenHands / Claude 3.7 Sonnet on full set)

**Multi-run evaluation.** To further assess the stability and potential of existing agents under repeated trials, we conduct a multi-run evaluation on the subset. Specifically, we re-run SWE-agent paired with GPT-4.1 for three independent runs under identical configurations (temperature = 0.6). As shown in Table 6, the resolved rates across the three runs are 15.33%, 16.67%, and 16.67%, yielding an average of 16.22%. When allowing up to three attempts per instance (Pass@3), the success rate increases to 21.67%. This suggests that sampling diversity (as reflected by Pass@k) can lead to certain performance gains, while the average Pass@1 remains relatively stable. In the following sections, we primarily discuss and compare methods based on their Pass@1 scores.

Table 6: Multi-run performance of SWE-agent + GPT-4.1

|  | Run 1 | Run 2 | Run 3 | Average | Pass3 |
|---|---|---|---|---|---|
| Resolved (%) | 15.33 | 16.67 | 16.67 | 16.22 | 21.67 |

### 4.3 Performance vs. Creation Date.

To investigate whether the recency of an issue affects its difficulty, we analyze the resolved rate across different creation periods. As shown in Figure 4, SWE-bench-Live includes a balanced distribution of instances across quarters from 2024Q1 to 2025Q1. The resolved rate, based on OpenHands with Claude 3.7 Sonnet on the full benchmark, remains relatively stable over time, fluctuating only modestly across quarters.

While there is a slight dip in resolved rate during 2024Q4, followed by a recovery in 2025Q1, the trend does not indicate a clear correlation between task recency and success rate. This suggests that newer issues are not inherently harder for current agents to solve, and that SWE-bench-Live maintains a consistent level of challenge across time. These results reinforce the benchmark's ability to deliver a steady and reliable evaluation signal, even as it continuously evolves with newly introduced instances.

### 4.4 Performance vs. Difficulty

We approximate the difficulty of a bug–fixing instance along two complementary axes. *Patch difficulty* is captured by the scope of the gold fix—the number of files it touches and the total lines modified—while *repository difficulty* is approximated by the overall size of the project in files and lines of code (LOC).

**Patch difficulty.** Figure 5 visualises resolved rate as a heat-map over patch scope. Success is high when the fix is local: a single-file patch that changes fewer than five lines is solved almost one time in two (48%). Performance degrades quickly as either dimension grows. Once the patch edits three or more files, or spans more than one hundred lines, the success rate falls below ten per-cent; patches that touch seven or more files are never solved. The sharp drop beyond the *one-file / few-lines* corner highlights a key limitation of current agents: they struggle to coordinate coherent edits across multiple files or to reason about large, intra-file changes.

**Repository difficulty.** Figure 7 in Appendix C plots resolved rate for every repository against its size (Python files on the x-axis, LOC on the y-axis). Bubble area reflects the number of instances drawn from each project, and red outlines mark the original SWE-bench repositories. A clear negative trend emerges: repositories with fewer than one hundred files and under twenty-thousand LOC often yield success rates above twenty per-cent, whereas projects exceeding five-hundred files rarely exceed five per-cent. Nevertheless, notable variance remains—some small-to-mid-size projects are still hard to fix, likely due to atypical build systems or complex domain logic—emphasising that size is an informative but imperfect proxy for difficulty.

Together, the two figures show that difficulty increases along both local (patch) and global (repository) dimensions, and that current code agents falter once fixes spill beyond a handful of lines or involve cross-file reasoning. Because SWE-bench-Live spans the full spectrum of these difficulty factors—while continuously adding fresh, unseen instances—it provides a stringent and up-to-date testbed for future advances in large-scale program repair.

## 5 Conclusion

We present SWE-bench-Live, the first continuously updating benchmark for evaluating large language models on real-world issue resolution tasks at the repository level for fresh issue fixing. By addressing key limitations of prior benchmarks such as dataset staleness, limited repository diversity, and manual curation cost, SWE-bench-Live provides a scalable, contamination resistant, and fully automated evaluation framework. At its core is REPOLAUNCH, an agent based pipeline that builds reproducible Docker environments and validates issue and pull request pairs through test execution, removing the need for manual intervention. Our empirical results across multiple agent and model combinations show that SWE-bench-Live presents significantly greater challenges than static datasets. The low resolution rates, especially on multi file patches and large codebases, highlight the limitations of current systems and the importance of live benchmarks in measuring true model generalization.

## Acknowledgment

We thank the reviewers for their invaluable feedback. We further thank Kenan Li and Rongzhi Li for extending RepoLaunch to multi-language and cross-platform support, and for their open-source contributions.

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

# A    Full Repositories List

| Type | Repository | License | #Instances | #Files | LoC |
|---|---|---|---|---|---|
| AI/ML | deepset-ai/haystack | Apache-2.0 | 64 | 433 | 84.8k |
| | instructlab/instructlab | Apache-2.0 | 52 | 142 | 28.2k |
| | keras-team/keras | Apache-2.0 | 48 | 900 | 249.7k |
| | kedro-org/kedro | Apache-2.0 | 27 | 179 | 40.4k |
| | pytorch/torchtune | BSD-3-Clause | 14 | 448 | 92.4k |
| | jupyterlab/jupyter-ai | BSD-3-Clause | 13 | 81 | 9.0k |
| | run-llama/llama_deploy | MIT | 12 | 216 | 15.1k |
| | stanfordnlp/dspy | MIT | 10 | 222 | 30.6k |
| | projectmesa/mesa | Apache-2.0 | 9 | 109 | 20.3k |
| | huggingface/smolagents | Apache-2.0 | 5 | 65 | 21.4k |
| | theOehrly/Fast-F1 | MIT | 4 | 92 | 20.7k |
| | cyclotruc/gitingest | MIT | 3 | 39 | 4.7k |
| | modelcontextprotocol/python-sdk | MIT | 2 | 114 | 13.4k |
| | camel-ai/camel | Apache-2.0 | 2 | 799 | 130.1k |
| | hiyouga/LLaMA-Factory | Apache-2.0 | 2 | 170 | 31.2k |
| | feast-dev/feast | Apache-2.0 | 2 | 673 | 103.3k |
| | openai/openai-agents-python | MIT | 1 | 212 | 29.8k |
| | huggingface/datasets | Apache-2.0 | 1 | 207 | 69.6k |
| | stanford-crfm/helm | Apache-2.0 | 1 | 891 | 122.1k |
| | freqtrade/freqtrade | GPL-3.0 | 1 | 458 | 130.1k |
| | lss233/kirara-ai | AGPL-3.0 | 1 | 261 | 25.5k |
| | arviz-devs/arviz | Apache-2.0 | 1 | 259 | 50.6k |
| | qubvel-org/segmentation_models.pytorch | MIT | 1 | 130 | 18.6k |
| | scikit-learn-contrib/category_encoders | BSD-3-Clause | 1 | 71 | 12.9k |
| | huggingface/open-r1 | Apache-2.0 | 1 | 29 | 4.0k |
| | gptme/gptme | MIT | 1 | 124 | 23.3k |
| DevOps | conan-io/conan | MIT | 136 | 1056 | 162.5k |
| | pylint-dev/pylint | GPL-2.0 | 57 | 2301 | 116.8k |
| | sphinx-doc/sphinx | N/A | 39 | 718 | 140.3k |
| | pdm-project/pdm | MIT | 34 | 221 | 32.3k |
| | beeware/briefcase | BSD-3-Clause | 24 | 508 | 89.3k |
| | bridgecrewio/checkov | Apache-2.0 | 21 | 4551 | 234.9k |
| | joke2k/faker | MIT | 20 | 754 | 351.4k |
| | python-attrs/attrs | MIT | 10 | 52 | 18.6k |
| | ipython/ipython | BSD-3-Clause | 10 | 293 | 79.8k |
| | koxudaxi/datamodel-code-generator | MIT | 10 | 599 | 60.3k |
| | tox-dev/tox | MIT | 7 | 225 | 23.8k |
| | dynaconf/dynaconf | MIT | 6 | 463 | 55.1k |
| | pypa/twine | Apache-2.0 | 6 | 34 | 6.6k |
| | wemake-services/wemake-python-styleguide | MIT | 6 | 396 | 52.5k |
| | Delgan/loguru | MIT | 6 | 168 | 19.2k |
| | kubernetes-client/python | Apache-2.0 | 3 | 783 | 267.2k |
| | olofk/fusesoc | BSD-2-Clause | 2 | 45 | 8.8k |
| | amoffat/sh | MIT | 2 | 5 | 7.4k |
| | facebookresearch/hydra | MIT | 2 | 439 | 41.4k |
| | home-assistant/supervisor | Apache-2.0 | 2 | 541 | 82.3k |
| | FreeOpcUa/opcua-asyncio | LGPL-3.0 | 1 | 168 | 344.4k |
| | pytest-dev/pytest | MIT | 1 | 260 | 99.5k |
| | iterative/dvc | Apache-2.0 | 1 | 554 | 85.3k |
| Web | reflex-dev/reflex | Apache-2.0 | 44 | 376 | 89.8k |
| | sissbruecker/linkding | MIT | 29 | 193 | 26.4k |
| | Kozea/WeasyPrint | BSD-3-Clause | 19 | 144 | 70.0k |
| | python-telegram-bot/python-telegram-bot | GPL-3.0 | 16 | 464 | 140.8k |
| | python-babel/babel | BSD-3-Clause | 11 | 75 | 23.1k |
| | falconry/falcon | Apache-2.0 | 11 | 262 | 58.5k |
| | aiogram/aiogram | MIT | 11 | 861 | 69.8k |
| | privacyidea/privacyidea | AGPL-3.0 | 10 | 483 | 167.5k |
| | urllib3/urllib3 | MIT | 10 | 81 | 31.3k |
| | ag2ai/faststream | Apache-2.0 | 6 | 1267 | 85.1k |
| | encode/starlette | BSD-3-Clause | 5 | 66 | 17.2k |
| | scrapinghub/dateparser | BSD-3-Clause | 2 | 274 | 67.1k |

| Type | Repository | License | #Instances | #Files | LoC |
|------|-----------|---------|-----------|--------|-----|
| Web | pallets/flask | BSD-3-Clause | 2 | 83 | 17.8k |
| | scrapy-plugins/scrapy-splash | BSD-3-Clause | 1 | 25 | 3.4k |
| | psf/requests | Apache-2.0 | 1 | 36 | 11.2k |
| | jpadilla/pyjwt | MIT | 1 | 26 | 6.9k |
| | slackapi/bolt-python | MIT | 1 | 562 | 60.8k |
| Database | pydata/xarray | Apache-2.0 | 29 | 226 | 179.2k |
| | geopandas/geopandas | BSD-3-Clause | 21 | 87 | 47.3k |
| | reata/sqllineage | MIT | 18 | 103 | 9.7k |
| | patroni/patroni | MIT | 17 | 117 | 45.9k |
| | piskvorky/smart_open | MIT | 6 | 64 | 12.4k |
| | wireservice/csvkit | MIT | 3 | 48 | 6.6k |
| | jazzband/tablib | MIT | 2 | 32 | 6.6k |
| | Flexget/Flexget | MIT | 2 | 657 | 108.6k |
| Scientific | pvlib/pvlib-python | BSD-3-Clause | 29 | 178 | 59.9k |
| | python-control/python-control | BSD-3-Clause | 15 | 155 | 70.7k |
| | mikedh/trimesh | MIT | 14 | 248 | 74.8k |
| | PyPSA/PyPSA | MIT | 10 | 129 | 32.3k |
| | shapely/shapely | BSD-3-Clause | 9 | 158 | 34.0k |
| | pybamm-team/PyBaMM | BSD-3-Clause | 6 | 581 | 113.4k |
| | beancount/beancount | GPL-2.0 | 2 | 194 | 48.0k |
| | sympy/sympy | N/A | 2 | 1574 | 760.5k |
| CLI | streamlink/streamlink | BSD-2-Clause | 39 | 510 | 84.0k |
| | beetbox/beets | MIT | 9 | 193 | 69.3k |
| | yt-dlp/yt-dlp | Unlicense | 5 | 1177 | 244.6k |
| | jarun/buku | GPL-3.0 | 2 | 27 | 7.1k |
| Misc | matplotlib/matplotlib | N/A | 85 | 904 | 263.6k |
| | fonttools/fonttools | MIT | 12 | 512 | 192.6k |
| | pytransitions/transitions | MIT | 2 | 39 | 12.4k |
| Cloud | aws-cloudformation/cfn-lint | MIT-0 | 102 | 2422 | 160.2k |
| | icloud-photos-downloader/icloud_photos_downloader | MIT | 4 | 73 | 15.5k |
| Desktop | qtile/qtile | MIT | 6 | 405 | 81.6k |
| | pwr-Solaar/Solaar | GPL-2.0 | 3 | 94 | 33.7k |

# B  Task Formulation

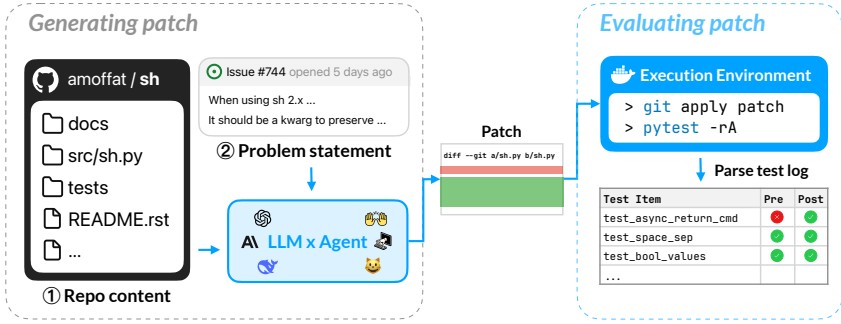

Figure 6: The issue-resolving task requires the model to generate a patch that addresses a given issue, with its correctness evaluated through test execution.

Issue resolving is the task that introduced by SWE-bench [10] for benchmarking AI coding capabilities. In simple terms, it simulates the process of a developer submitting a pull request to address an issue. The formulation of the issue-resolving task is illustrated in Figure 6.

**Generating Patch.** The task input includes the problem statement of the issue, which is the description written by the issue reporter, as well as a snapshot of the codebase at the time the issue was filed (obtained by resetting to the `base_commit`). The model has access to full content of the codebase, after then it is tasked with generating a patch that fixes the given issue, analogous to the file changes submitted in a pull request. In practice, the expected output is in the `.diff` format.

**Evaluating Patch.** Once a patch is proposed by the model, we assess its correctness by applying it to the target codebase and executing the repository's test suite. The output of the test execution are parsed using a log parser function, which extracts the status of each individual test case. These results are then compared against the expected test case transitions pre-defined for the issue, specifically FAIL_TO_PASS and PASS_TO_PASS. FAIL_TO_PASS refers to test cases that originally failed prior to patch application—typically those introduced in the corresponding pull request—and are expected to pass if the proposed solution is correct. A correct patch should successfully cause these failing tests to pass, without causing regressions in the already passing tests.

## C  Performance vs Repository difficulty

The following Figure 7 plots resolved rate for every repository against its size (Python files on the x-axis, LOC on the y-axis). For detailed interpreatation of the figure, please see Section 4.4.

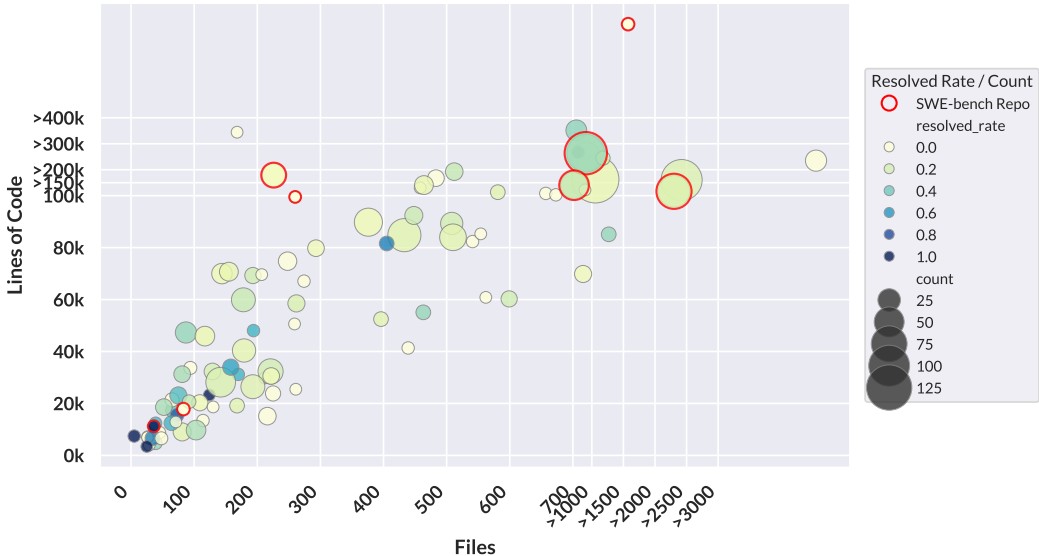

Figure 7: Resolved rate in relation to the number of files and lines of code of a repository.

## D  Dataset Fields

Table 7 provides a detailed description of the fields included in the SWE-bench-Live dataset, along with how they are obtained during the curation process.

## E  Experimental Setup Details

In this section, we present additional details of the experimental setup to facilitate reproducibility.

**Hyperparameters used in the experiments.** For OpenHands, we set a maximum of 60 iterations per instance, with the LLM configured to use a temperature of 0.0 and a top-p value of 1.0 as default. For SWE-agent, we limit the number of LLM calls to 100 per instance, with the temperature set to 0.0 and top-p to 1.0. For Agentless, both the number of localization samples and repair samples are set to 1, corresponding to a single rollout. The LLM temperature is set to 0.8 during the localization phase, as defined by the agent's default, and 0.0 for all other phases. In our experiments, we omit the regression test-based reranking stage of Agentless, retaining only the localization and repair stages. The LLM calls within REPOLAUNCH are configured with a temperature of 0.0.

**Random seed in subset splitting.** The only stochastic component in this work arises during the sampling of the lite subset, where we set the random seed to 42.

Table 7: The required fields for a typical issue-solving task instance. Fields marked with * are **newly added** in SWE-bench-Live compared to SWE-bench.

| Field | Type | Description |
|---|---|---|
| base_commit | str | The commit on which the pull request is based, representing the repository state before the issue is resolved. |
| patch | str | Gold patch proposed by the pull request, in .diff format. |
| test_patch | str | Modifications to the test suite proposed by the pull request that are typically used to check whether the issue has been resolved. |
| problem_statement | str | Issue description text, typically describing the bug or requested feature, used as the task problem statement. |
| FAIL_TO_PASS | List[str] | Test cases that are expected to successfully transition from failing to passing are used to evaluate the correctness of the patch. |
| PASS_TO_PASS | List[str] | Test cases that are already passing prior to applying the gold patch. A correct patch shouldn't introduce regression failures in these tests. |
| *image_key | str | Instance-level docker image that provides an execution environment. |
| *test_cmds | List[str] | The command(s) used to run the test suite is identified by the verify agent in REPOLAUNCH. It is required to enable detailed logging of each test item's status (e.g., by using the pytest -rA option). |
| *log_parser | str | The type of log parser required for the instance—by default, pytest. |

**Computational resources.** All LLM calls in this work are made through official APIs. The experiments involve parallel execution of multiple Docker containers for test execution. We conduct all the experiments on a CPU server equipped with an Intel Xeon Gold 6338 @ 2.00GHz (128 cores) and 2TB of RAM.

# F  SWE-bench-Live Verified: Automatic Quality Filtering

We introduce **SWE-bench-Live-Verified**, a high-quality subset of SWE-bench-Live automatically constructed via LLM-based filtering to ensure task validity and evaluation reliability.

**Method.** Each instance from SWE-bench-Live is assessed by an LLM given its issue description, gold patch, and FAIL_TO_PASS tests. The model classifies instances into eight categories reflecting common quality issues, including vague or misleading issues, underspecified tests, trivial fixes, and environmental failures. Only those judged as **well-posed and evaluable** are retained.

**Filtering Accuracy.** Applying o3 to 1,699 SWE-bench-Full instances, we obtain 72% precision and 40% recall against human-labeled SWE-bench-Verified, or 92% precision and 35% recall when excluding trivial cases. GPT-4.1 yields lower recall (10%) and slightly lower precision (76–86%), showing that reasoning-specialized models better identify valid debugging tasks. The initial SWE-bench-Live Verified release contains 500 instances (38% of full set) from July 2024 – April 2025.

**Agent Evaluation.** Table 8 summarizes model performance on this verified subset. Compared with the unfiltered set, these methods achieve slightly higher scores on the verified set, suggesting that the filtering process removes a certain amount of invalid or ill-posed tasks, thereby improving the robustness of the evaluation.

Table 8: Performance (%) on SWE-bench-Live Verified.

| Agent | GPT-4o | GPT-4.1 | Claude 3.7 Sonnet | DeepSeek-V3 |
|---|---|---|---|---|
| SWE-agent | 14.94 | 16.09 | 19.54 | 13.22 |
| s-Agentless | 13.22 | 11.49 | 12.07 | 13.79 |
| OpenHands | 6.32 | 12.07 | 20.69 | 13.22 |

# G  Limitations

**Randomness caused by LLMs**: We use the LLMs as the core engine to conduct all the experiments, which might lead to potential randomness caused by different LLM calls. Since the experiments

require extensive LLM calls while the overall budget is limited, we do not repeat the experiments for multiple times. To reduce the randomness, we control the execution environment to be the same and set the temperature and top_p to zero.

**Language limitation**: Our benchmark SWE-bench-Live primarily focuses on the Python language only, which might be limited. Since our key contribution is to propose a live benchmark with an automated and scalable method, we follow the same language choice as existing benchmarks like SWE-bench. In the future, we plan to extend SWE-bench-Live to multiple languages such as Java, Go, and etc.

# H   Prompts in REPOLAUNCH

---

**Prompt for Relevant Files Identification**

```
Given this repository structure:
--- BEGIN REPOSITORY STRUCTURE ---
{structure}
--- END REPOSITORY STRUCTURE ---
List the most relevant files for setting up a development environment,
including:
0.  CI/CD configuration files
1.  README files
2.  Documentation
3.  Installation guides
4.  Development setup guides
Format each file with its relative path (relative to project root) to
be wrapped with tag <file> </file>, one per line.
```

---

**Prompt for Base Image Selection**

```
Based on related file:
{related_files}
Please recommend a suitable base Docker image.  Consider:

1.  The programming language and version requirements
2.  Common system dependencies
3.  Use official images when possible

Select a base image from the following candidate list:
{candidate_images}
Wrap the image name in a block like <image>python:3.11</image> to
indicate your choice.
```

---

## Prompt for Setup Agent

You are a developer.  Your task is to install dependencies and set up
a environment that is able to run the tests of the project.

- You start with an initial Docker container based on {base_image}.
- You interact with a Bash session inside this container.
- Project files are located in /testbed within the container, and your
current working directory of bash is already set to /testbed.
- No need to clone the project again.

The final objective is to successfully run the tests of the project.
### Attention:
- For Python project, you should make sure the package is installed
from source in the editable mode before running tests (for example
'pip install -e .')  to have a development environment.
- For Python project, avoid use tox to run test if possible as it is
designed specifically for CI. Read tox.ini file to find how to setup
and run the test.
You run in a loop of Thought, Action, Observation.  At the end of the
loop you should use Action to stop the loop.
Use Thought to describe your thoughts about the question you have
been asked.
Use Action to run one of the actions available to you.
Observation will be the result of running those actions.
> Important Note:  Each step, reply with only **one** (Thought,
Action) pair.
> Important Note:  Do not reply **Observation**, it will be provided
by the system.
Your available actions are:
{tools}
Observation will be the result of running those actions.

Project Structure:  the structure of the project, including files and
directories.
Related Files:  the content of related files of the project that may
help you understand the project.
Thought:  you should always think about what to do
Action:  decide an action to take
Observation:  the result of the action

...  (this Thought/Action/Observation can repeat N times) ...

Thought:  I think the setup should be fine
Action:  stop the setup
Answer:  the final result

Begin
Project Structure:  {project_structure}
Related Files:  {docs}

You are a developer. Your task is to verify whether the environment
for the given project is set up correctly. Your colleague has set up
a Docker environment for the project. You need to verify if it can
successfully run the tests of the project.

- You interact with a Bash session inside this container.
- The container is based on {base_image}.
- The setup commands that your colleague has run are {setup_commands}
- Project files are located in /testbed within the container, and your
current working directory of bash is already set to /testbed.
- Use the same test framework as your colleague, because that aligns
with the setup stage.
- Only test commands, skip linting/packaging/publishing commands.
- Do not change the state of the environment, your task is to verify
not to fix it. If you see issues, report it not fix it.
- You can tolerate a few test cases failures-as long as most tests
pass, it's good enough.

## Important Note:

Your test command must output detailed pass/fail status for each
test item. This is mandatory. For example, with pytest, use the
-rA option to get output like:

```
PASSED tests/test_resources.py::test_fetch_centromeres
PASSED tests/test_vis.py::test_to_ucsc_colorstring
```

Since we need to parse the test output to extract a test item →
status mapping, **this requirement is mandatory**. If you observed
that your test command does not produce such detailed output, you
must adjust it accordingly.

In summary, your goal is:
1. Write the test commands that could output detailed pass/fail
status for each test item, you can iterate until it does. (this is
mandatory, DO NOT ignore this requirement!!! This is your obligation
to correctly identify the test commands to run the test suite of the
project, and find a way to output detailed pass/fail status)
2. Run the test command to verify if the environment is set up
correctly. If not, report any observed issues. If you think the
setup is correct, report none issue.

