# OpenReview forum: "SWE-bench Goes Live!"
_NeurIPS.cc/2025/Datasets_and_Benchmarks_Track — NeurIPS 2025 Datasets and Benchmarks Track poster_

### Official Review · Reviewer_HPe4 · 2025-06-30

**Rating:** 5
**Confidence:** 5

**Summary:**

SWE-bench is one of the most popular benchmarks for judging LM's coding abilities, but it is a static benchmark. This contribution is about updating it on a rolling basis, so that data leakage does not affect model's scores. They also spent a lot of effort to make the SWE-bench task collection pipeline automated, which will be widely useful far beyond just this benchmark. Now when anyone wants to build any type of SWE-benchmark (for example, let's say they wanna do software issues with images in Rust) they will have a much easier time. I believe this paper should be accepted.

**Additional Feedback:**

1. see question in weaknesses
2. how did having more repos help? what can we learn from the large number of repos that we couldn't learn if we had just 10 repos in this benchmark?

**Dataset Code Accessibility:**

Yes

**Ethical Considerations:**

No, there are no or only very minor ethics concerns

**Final Justification:**

This is an interesting and useful paper and artifact and I would like to see it get accepted.

**Limitations Weaknesses:**

1. While the *absolute scores* for each system is much lower than it is on SWE-bench, the rankings of the different agents+models seem to be somewhat similar to the rankings on SWE-bench. What can we learn from this benchmark that we couldn't learn from SWE-bench?

**Strengths Contributions:**

1. Live version of SWE-bench with many new issues, that will be constantly updated.
2. Wide repo coverage: 90+, instead of the 12 that OG swe-bench had.
3. Fully-automated pipeline for crawling swe-bench-like tasks: this will save many people a lot of time in the future.
4. Strong suite of baselines: SWE-agent, Agentless, and OpenHands.

---

> ### Author Rebuttal · Authors · 2025-07-31
>
> We sincerely thank you for your positive review and valuable feedback! We have carefully considered your comments and respond to each of them below.
>
> > While the absolute scores for each system is much lower than it is on SWE-bench, the rankings of the different agents+models seem to be somewhat similar to the rankings on SWE-bench. What can we learn from this benchmark that we couldn't learn from SWE-bench?
>
> Thank you for this insightful question. While it is true that the relative rankings of agents and models remain largely consistent between SWE-bench and SWE-bench-Live, our benchmark offers several unique insights and advantages:
>
> - SWE-bench-Live expands repository coverage by nearly an order of magnitude and continuously incorporates the latest issues. This greatly increases the diversity, freshness, and realism of evaluation scenarios. As a result, it enables us to assess how well agents and models generalize to truly novel, previously unseen projects and problem types—something not possible with a static benchmark.
>
> - The substantially lower absolute scores observed on SWE-bench-Live suggest that prior benchmarks may be subject to overfitting or even data contamination. SWE-bench-Live’s regularly updated set of tasks provides a much more stringent and contamination-free testbed for measuring model performance.
>
> The fact that rankings or relative performance are consistent with SWE-bench further reinforces that our benchmark can objectively reflect models’ issue-resolving abilities in a mutually validating way.
>
> In summary, SWE-bench-Live enables the community to track both absolute improvements and generalization in model capabilities, while mitigating the risks of overfitting and staleness inherent in previous static benchmarks. We believe this is essential for robust and reliable evaluation of LLMs.
>
> > how did having more repos help? what can we learn from the large number of repos that we couldn't learn if we had just 10 repos in this benchmark?
>
> Expanding the benchmark to a broader set of repositories offers several key advantages:
>
> - **Reduced Overfitting and Data Contamination:** A small, static benchmark risks overfitting and memorization. In contrast, a larger and continuously updated benchmark like SWE-bench-Live makes it much harder for models to exploit quirks of a fixed dataset, resulting in a more reliable measure of generalization.
> - **Capturing Current Software Engineering Practices:** By regularly incorporating new repositories and live GitHub issues, SWE-bench-Live reflects the latest development tools and frameworks, ensuring the evaluation remains realistic and up to date.
> - **Greater Coverage and Diversity:** A larger pool spans more domains, codebase sizes, and architectures, enabling a thorough assessment of model strengths and weaknesses across real-world scenarios (see Figure 2, 7).
>
> Our experimental results further show that methods achieving over 60% accuracy on SWE-bench saw a dramatic performance drop on SWE-bench-Live. This highlights the importance of evaluating on a wider, more current set of repositories.

---

### Official Review · Reviewer_J8NV · 2025-07-01

**Rating:** 5
**Confidence:** 3

**Summary:**

This paper introduces SWE-bench-Live, a continuously updating benchmark for evaluating large language models on real-world software issue resolution tasks. The work addresses key limitations of existing static benchmarks like SWE-bench, including dataset staleness, limited repository coverage, and reliance on manual curation. The authors propose REPOLAUNCH, a fully automated pipeline that constructs benchmark instances by crawling GitHub issues, setting up execution environments via Docker, and validating task instances through test execution. The initial release contains 1,319 tasks from 93 Python repositories with issues created since 2024. Experimental evaluation reveals significantly lower performance on SWE-bench-Live compared to static benchmarks, suggesting potential overfitting to existing datasets.

**Dataset Code Accessibility:**

Yes

**Ethical Considerations:**

No, there are no or only very minor ethics concerns

**Final Justification:**

The discussion has clarified some of my concerns to a certain extent, so I choose to maintain my original score (5/Accpect) unchanged.

**Limitations Weaknesses:**

1. The authors acknowledge omitting statistical significance testing due to budget constraints, but this makes it difficult to assess the reliability of performance differences. The decision to exclude reranking from Agentless evaluation, while justified by infrastructure constraints, may not reflect the agent's full capabilities.

2. While the paper identifies large performance differences between benchmarks, it provides limited investigation into the underlying causes. More detailed error analysis, qualitative examination of failed cases, or comparison of task characteristics could provide deeper insights into why models struggle more on live data.
3. The restriction to Python repositories, while understandable for initial release, limits the benchmark's generalizability. Modern software development is increasingly polyglot, and issue resolution capabilities should be evaluated across multiple programming languages.

**Strengths Contributions:**

- The paper is well-organized with informative figures (particularly Figure 1 showing the pipeline) and comprehensive experimental details. The writing is clear and the motivation is well-established.
- The REPOLAUNCH framework automates the historically manual and labor-intensive process of benchmark construction. The three-stage pipeline (issue crawling, environment setup, validation) is well-designed and addresses scalability bottlenecks that have limited previous benchmarks.

- The work tackles fundamental problems with static benchmarks, particularly data contamination risks and limited diversity. The focus on post-2024 issues provides a contamination-resistant evaluation setting that better reflects real-world software development.

- The evaluation is thorough, testing multiple agent frameworks (OpenHands, SWE-Agent, Agentless) with four state-of-the-art LLMs across different metrics (resolved rate, patch apply rate, localization success). The controlled comparison between SWE-bench and SWE-bench-Live provides valuable insights. dramatic performance difference between static and live benchmarks (43.20% vs 19.25% for the best configuration) reveals potential overfitting issues and highlights the importance of continuous evaluation. The analysis showing better performance on repositories originally in SWE-bench further supports this concern.

---

> ### Author Rebuttal · Authors · 2025-07-31
>
> We sincerely thank you for your positive review and valuable feedback! We have carefully considered your comments and respond to each of them below.
>
> > The authors acknowledge omitting statistical significance testing due to budget constraints, but this makes it difficult to assess the reliability of performance differences. The decision to exclude reranking from Agentless evaluation, while justified by infrastructure constraints, may not reflect the agent's full capabilities.
>
> Thank you for your thoughtful feedback. As we reply to Reviewer cdHm, implementing the reranking in Agentless typically involves generating and evaluating hundreds of candidate patches for each instance—a process that is both computationally expensive and requires substantial modifications to the infrastructure. Given the scope and scale of our benchmark, we adopted the “pass@1” evaluation protocol consistent with prior works such as Multi-SWE-bench [1]. We very much encourage the original authors and the broader community to submit results using alternative or enhanced evaluation settings on SWE-bench-Live.
>
> Additionally, we now have the evaluation results for the latest models (OpenHands+CodeAct, evaluated on the lite subset):
>
> | Model | % Resolved |
> |-------|-------|
> | Claude 4 Sonnet | 27.7 |
> | Qwen3-Coder | 26.3 |
> | Kimi-K2-Instruct | 22.3 |
>
> [1] Zan, Daoguang, et al. "Multi-swe-bench: A multilingual benchmark for issue resolving." arXiv preprint
>
> > While the paper identifies large performance differences between benchmarks, it provides limited investigation into the underlying causes. More detailed error analysis, qualitative examination of failed cases, or comparison of task characteristics could provide deeper insights into why models struggle more on live data.
>
> Thank you for highlighting the importance of deeper error analysis to understand the performance gap between benchmarks. In our work, we have  included several key investigations:
>
> - In Section 4.2, we categorized instances by repository origin, showing that models achieve 22.96% success on SWE-bench repositories versus 18.89% on new repositories. This indicates a degree of overfitting to familiar codebases and underscores the importance of broader repository coverage for objective evaluation.
> - To further investigate the reason, we conduct additional analysis on the task complexity. We find that the median oracle patch in SWE-bench-Live involves **2 files and 24 lines,** compared to SWE-bench's average of **1 file and 12 lines**—demonstrating that SWE-bench-Live presents more complex challenges.
>
> We believe these factors explain much of the observed performance gap. For the revision, we will include a more comprehensive taxonomy of failure modes and qualitative case studies. Additionally, we have made all experimental logs and agent trajectories publicly available to support further community-driven analysis.
>
> While we plan to expand this analysis, our current findings already provide compelling evidence that live evaluation introduces fundamentally new and harder challenges compared to static benchmarks, further motivating the need for SWE-bench-Live.
>
> > The restriction to Python repositories, while understandable for initial release, limits the benchmark's generalizability. Modern software development is increasingly polyglot, and issue resolution capabilities should be evaluated across multiple programming languages.
>
> Thank you for your valuable feedback. As we reply to Reviewer cdHm, expanding the language scope has been a key priority for us. In response, we have already extended the RepoLaunch to support JavaScript/TypeScript, Rust, Java, and Go. We plan to update SWE-bench-Live to include more repositories in other programming languages. This enhancement will allow us to include a more diverse range of repositories and enable evaluation of agent and model generalization across multiple programming languages.

---

> > ### Comment · Reviewer_J8NV · 2025-08-05
> >
> > Thank you for the author's response. This has clarified some of my concerns to a certain extent, so I choose to maintain my original score unchanged.

---

### Official Review · Reviewer_eueD · 2025-07-02

**Rating:** 5
**Confidence:** 4

**Summary:**

SWE-bench-Live is a live-updating, contamination-resistant benchmark for evaluating LLMs and code agents on real-world issue resolution. Unlike previous static datasets, it uses a fully automated pipeline, REPOLAUNCH, to continuously gather, set up, and validate new bug-fixing tasks from 93 active Python repositories. Each task is provided with a reproducible Docker environment. Empirical results reveal that even the most advanced models and agents perform much worse on SWE-bench-Live than on prior benchmarks, exposing overfitting and major gaps in generalization.

**Dataset Code Accessibility:**

Yes

**Ethical Considerations:**

No, there are no or only very minor ethics concerns

**Final Justification:**

The authors have provided point-to-point responses to my concerns, they are either partially addressed with new results (Q1 with results from additional programming languages) or clarified (Q2, Q3). I maintain my recommendation as SWE-Bench Live gaugea important real-world agentic SDE skills and is less prone to saturation.

**Limitations Weaknesses:**

- Currently only covers Python repositories; does not assess other programming languages.
-  Task instance validation relies on test suite transition info (fail-to-pass, pass-to-pass), which may miss certain bugs or overfit to existing test coverage.
- REPOLAUNCH limitations:

       1. If some repositories or commits may have undocumented system dependencies, nonstandard build steps, or platform-specific quirks that REPOLAUNCH may miss, wouldn't it lead to setup failures?
       2. no robustness study. Automated identification of relevant files (e.g., setup scripts, CI configs) can miss edge cases or important files not following common conventions. Also there is no human-in-the-loop recovery mechanism, there is no fallback to manual debugging when automated setup fails, so some potentially solvable instances may be discarded.

**Strengths Contributions:**

- SWE-Bench-live is the first live-updating benchmark for for complex, repository-level tasks to evaluation LLM-based bug-fixing, with the advantage of minimizing data leakage and remaining relevant as codebases and LLMs evolve. The coverage spans a wide spectrum of domains (AI/ML, DevOps, Web, Database, etc.), enhancing generalizability of benchmark results.
- Scalability:

        1. Fully automated pipeline (REPOLAUNCH) for issue/PR mining, Docker-based environment setup, and test validation removes the manual bottleneck.
        2. Includes 1,319 tasks from 93 real-world repositories (as of April 2025), far surpassing prior benchmarks’ coverage and diversity.

- Reproducibility: Each task has a dedicated Docker image for reproducible LLM and code agent evaluations.
- All data, code, and environments are publicly available.

---

> ### Author Rebuttal · Authors · 2025-07-31
>
> We sincerely thank you for your positive review and valuable feedback! We have carefully considered your comments and respond to each of them below.
>
> > Currently only covers Python repositories; does not assess other programming languages.
>
> Thank you for your valuable feedback. As we reply to Reviewer cdHm, expanding the language scope has been a key priority for us. In response, we have already extended the RepoLaunch to support JavaScript/TypeScript, Rust, Java, and Go. We plan to update SWE-bench-Live to include more repositories in other programming languages. This enhancement will allow us to include a more diverse range of repositories and enable evaluation of agent and model generalization across multiple programming languages.
>
> We also conducted some preliminary experiments to explore the effectiveness of RepoLaunch on repositories in other programming languages, with the following results:
>
> | Language   | Repo Suc Rate | Task Suc Rate | Avg. duration (mins) |
> |------------|---------------|---------------|----------------------|
> | Javascript | 19/20         | 71/100        | 9.5                  |
> | Rust       | 15/20         | 61/100        | 11.0                 |
> | Go         | 13/20         | 33/100        | 11.1                 |
>
> With RepoLaunch, we are actively working on expanding the scope and language coverage of SWE-bench-Live, continuously providing fresh issue-resolving tasks with environments for the research community.
>
> > Task instance validation relies on test suite transition info (fail-to-pass, pass-to-pass), which may miss certain bugs or overfit to existing test coverage.
>
> Thank you for raising this concern. The evaluation of issue-resolving tasks like SWE-bench is largely test-driven, and the quality of the tests substantially affects the reliability of a benchmark instance. We have used an LLM-based quality screening pipeline to create a verified subset, similar to what SWE-bench verified did, to ensure as much as possible that the tests used to check whether issues are successfully resolved are high-quality - tests that won't be too narrow while being able to cover as many correct solutions as possible. Based on this, we created a Verified subset of SWE-bench-Live containing 500 high-quality issue-resolving tasks from the original 1699 tasks. We will include the quality screening workflow and analysis in the paper revision. We believe it can greatly enhance the reliability of evaluation!
>
> > If some repositories or commits may have undocumented system dependencies, nonstandard build steps, or platform-specific quirks that REPOLAUNCH may miss, wouldn't it lead to setup failures?
>
> Thank you for raising this important concern. RepoLaunch is designed to emulate the standard workflow that human developers follow when setting up unfamiliar projects: it scans and processes documentation files (such as README, requirements files, and setup scripts), identifies relevant dependencies, and executes standard build and test commands. When essential steps or dependencies are missing from the repository’s documentation, this poses a significant challenge not only for automated agents, but also for human users attempting to build or test the code. In such cases, setup failures are indeed expected and reflect the real-world reproducibility challenges often encountered in open-source software.
>
> At the same time, our automated pipeline naturally selects for repositories with clearer documentation and more robust build and test processes. Consequently, the benchmark emphasizes higher-quality projects that are both more maintainable and more representative of best practices within the software engineering community.
>
> Finally, according to our experiments, RepoLaunch successfully set up 379 out of 536 Python repositories—demonstrating a strong success rate without any manual intervention and highlighting its reliability as an automated build approach.
>
> > No robustness study. Automated identification of relevant files (e.g., setup scripts, CI configs) can miss edge cases or important files not following common conventions. Also there is no human-in-the-loop recovery mechanism, there is no fallback to manual debugging when automated setup fails, so some potentially solvable instances may be discarded.
>
> Thank you for raising this important point. The core motivation behind designing a fully automated environment setup pipeline is to enable large-scale, reproducible, and continuously updatable benchmarking without incurring significant manual effort. We recognize that a purely automated approach will not achieve the same success rate as an expert human developer. Nonetheless, our pipeline still achieved a good success rate—successfully setting up 379 out of 536 Python repositories, which reflects a practical balance between scalability and coverage.
>
> While we trade off some coverage for automation and scalability, we believe this design choice is necessary to achieve the goals of SWE-bench-Live. We are actively exploring future directions, such as incorporating lightweight human verification or more advanced heuristics to further enhance robustness while maintaining efficiency.

---

> > ### Comment · Reviewer_eueD · 2025-08-08
> >
> > Thank you for your clarifications. My concerns are addressed and I will maintain my recommendation.

---

> ### Comment · Area_Chair_ExoL · 2025-08-06
> **AC Message: Please join the discussion ASAP**
>
> Hi reviewer eueD,
>
> As required by NeurIPS, please read the authors' rebuttal and actively participate in the discussion ASAP.
>
> Thank you,
>
> Your AC

---

### Official Review · Reviewer_cdHm · 2025-07-21

**Rating:** 5
**Confidence:** 4

**Summary:**

This paper introduces SWE‑bench‑Live, the first continuously updatable benchmark for real‐world GitHub issue resolution at the repository level. It addresses three key limitations of existing benchmarks (e.g., static snapshots, narrow repository coverage, extensive manual effort) by:

Automated Pipeline (REPOLAUNCH): An LLM‑driven agentic workflow that identifies relevant setup files, selects a base Docker image, iteratively installs dependencies via a ReAct loop, and verifies the test suite, all without manual intervention (Sec. 3.2).

Live Dataset: A dataset of 1,319 issue‑PR tasks from 93 popular Python repositories (issues created Jan 2024–Apr 2025), each packaged in a reproducible Docker image (Sec. 3.4).

Comprehensive Evaluation: Benchmarking three agent frameworks (OpenHands, SWE‑Agent, Agentless) with four state‑of‑the‑art LLMs (GPT‑4o, GPT‑4.1, Claude 3.7 Sonnet, DeepSeek V3) on both a Lite subset and the full dataset. Results show a dramatic drop in “Resolved Rate” (max 19.3%) compared to static SWE‑bench Verified (>60%), highlighting over‑fitting in prior benchmarks (Sec. 4.2–4.3).

**Dataset Code Accessibility:**

Yes

**Ethical Considerations:**

No, there are no or only very minor ethics concerns

**Final Justification:**

After considering the authors’ rebuttal and discussion, my final score remains 5: Accept

**Limitations Weaknesses:**

Language & Domain Scope

Python‑Only: Restricts applicability to the broader software ecosystem. Extending to Java, Go, etc., would enhance generality (Sec. F).

Statistical Robustness

Single‐Run Evaluations: All metrics (Resolved Rate, Apply Rate, Loc. Suc. Rate) are reported from one run with temperature 0.0. Lack of confidence intervals or repeated trials may mask variance. Including multi‑seed experiments and statistical tests would strengthen conclusions.

Evaluation Metrics

Limited Quality Measures: Focuses solely on pass/fail outcomes. Metrics such as patch correctness beyond tests (e.g., static code analysis), runtime overhead, or human‐readability could offer a more holistic assessment.

Pipeline Reliability Details

Failure Case Reporting: The paper does not quantify REPOLAUNCH’s success/failure rate on all candidate instances, nor categorize common failure modes. Providing these statistics (e.g., % of repos automatically setup, top failure causes) would help gauge pipeline robustness.

Agent Comparisons

Incomplete Agentless Evaluation: Omitting reranking in the Agentless pipeline may understate its capabilities. Clarifying this decision’s impact or integrating a lightweight reranking heuristic could yield fairer comparisons.

Actionable Suggestions

Extend Language Support: Prototype REPOLAUNCH on a non‑Python codebase to demonstrate generality.

Enhance Statistical Analysis: Report results over multiple seeds/temperatures and include standard deviations or confidence intervals.

Enrich Metrics: Incorporate patch quality scores (e.g., CodeBLEU), resource usage, and build times.

Document Pipeline Outcomes: Publish a breakdown of environment setup success rates and error taxonomy.

Complete Agent Pipelines: Integrate or simulate the Agentless reranking stage to ensure apples‑to‑apples comparisons.

**Strengths Contributions:**

First Live-Updating Issue‐Resolution Benchmark: Unlike static datasets (SWE‑bench, Multi‑SWE‑bench) or algorithmic live benchmarks (LiveCodeBench), SWE‑bench‑Live continuously streams real PRs, preventing data contamination and reflecting current software challenges (Sec. 1, Fig. 1).

Fully Automated, Scalable Pipeline: REPOLAUNCH eliminates months of manual curation by automating environment setup and test validation for each snapshot, enabling monthly updates (Sec. 3.2, Fig. 1).

Technical Rigor

LLM‑Driven ReAct Workflow: Integrates “Thought→Action→Observation” loops to handle dependency installation and build steps, and a “time‑machine” PyPI proxy to enforce historical package versions (Sec. 3.2).

Validation Mechanism: Implements multi‑run test execution and strict FAIL_TO_PASS / PASS_TO_PASS criteria to ensure patches truly resolve the target issue (Sec. 3.3).

Empirical Breadth

Diverse Agent & Model Comparison: Evaluates three representative agent frameworks across four top‐tier LLMs, on both Lite (300 instances) and full (1,319 instances) sets (Sec. 4.1, Tables 3–4).

Difficulty Analysis: Correlates solved rates with patch size (files/hunks/lines) and repository scale (LOC/files), revealing clear failure modes when fixes span multiple files or large codebases (Fig. 5, App. C).

Presentation & Reproducibility

Clear Organization: Logical structure—from motivation and related work to methods, experiments, and limitations.

Rich Supplemental Material: Detailed appendices on prompts (App. G), data fields (App. D), hyperparameters, and resource specs facilitate replication.

Open Source Release: Dataset, Docker images, and code are publicly available, lowering barriers for future research.

---

> ### Author Rebuttal · Authors · 2025-07-31
>
> We sincerely thank you for your positive review and valuable feedback! We have carefully considered your comments and respond to each of them below.
>
> > Restricts applicability to the broader software ecosystem. Extending to Java, Go, etc., would enhance generality (Sec. F)
>
> Thank you for your valuable feedback regarding the language coverage of SWE-bench-Live. Yes, we now support multi-language! We have further adapted RepoLaunch to automatically set up execution environments for repositories in a broader range of languages. It currently supports JavaScript/TypeScript, Rust, Java, and Go. We will update SWE-bench-Live to include repositories in other programming languages, enabling us to evaluate the generalization capabilities of agents and models. We believe these forthcoming updates will address your concern and make SWE-bench-Live a more broadly applicable and valuable resource for the research community.
>
> > All metrics (Resolved Rate, Apply Rate, Loc. Suc. Rate) are reported from one run with temperature 0.0. Lack of confidence intervals or repeated trials may mask variance. Including multi‑seed experiments and statistical tests would strengthen conclusions.
>
> Thanks for the suggestion. The evaluation in the paper was essentially a “one-run” setting. However, agents like SWE-agent and OpenHands may internally perform multiple trials, such as generating their own reproduce tests and only submitting a result after repeated trials.
>
> Our current evaluation reports results from a single run. This design decision was primarily motivated by computational constraints, as running large-scale experiments with multiple seeds or repeated trials would significantly increase the required resources. We also followed the established evaluation protocol used in prior works [1] that similarly reports “pass@1” scores based on a single run. We hope model vendors or the community can take on this task if they want to heavily test their models or agents.
>
> > Focuses solely on pass/fail outcomes. Metrics such as patch correctness beyond tests (e.g., static code analysis), runtime overhead, or human‐readability could offer a more holistic assessment.
>
> Thank you for this insightful suggestion regarding the inclusion of additional evaluation metrics beyond pass/fail test outcomes.
> We agree that relying solely on test-driven outcomes may not fully capture all aspects of patch quality. In our evaluation in the original paper (see Tables 3–5, Section 4), we have taken a step toward a more holistic assessment by including two supplementary metrics: Patch Apply Rate, which measures the syntactic correctness of generated patches, and Localization Success Rate, which evaluates whether the modified files match the gold patch at the file level. These metrics aim to provide further insight into agent and model performance beyond just test outcomes.
>
> > The paper does not quantify REPOLAUNCH’s success/failure rate on all candidate instances, nor categorize common failure modes. Providing these statistics (e.g., % of repos automatically setup, top failure causes) would help gauge pipeline robustness.
>
> According to our experiments at the time, the overall repository success rate was 379 out of 536. For SWE-bench repositories specifically, RepoLaunch achieved a 12/12 success rate in setting up the environments.
>
> Additionally, we conducted preliminary experiments to evaluate RepoLaunch’s performance on repositories in other programming languages (using GPT-4.1):
>
> | Language   | Repo Suc Rate | Task Suc Rate | Avg. duration (mins) |
> |------------|---------------|---------------|----------------------|
> | Javascript | 19/20         | 71/100        | 9.5                  |
> | Rust       | 15/20         | 61/100        | 11.0                 |
> | Go         | 13/20         | 33/100        | 11.1                 |
>
> We acknowledge that understanding common failure patterns of RepoLaunch is important. The most frequent causes of setup failure included: (1) incomplete or ambiguous dependency specifications in the repository, (2) incompatibilities due to version drift in dependencies, and (3) project-specific build or test procedures not documented in standard files. We are actively working to further categorize and mitigate these failure cases and will provide an expanded breakdown in the final version.
>
> > Omitting reranking in the Agentless pipeline may understate its capabilities. Clarifying this decision’s impact or integrating a lightweight reranking heuristic could yield fairer comparisons.
>
> Thank you for raising this important point regarding reranking in the Agentless pipeline.
>
> We agree that reranking can significantly impact the reported performance of Agentless. However, implementing full reranking in the Agentless pipeline typically requires generating and evaluating hundreds of candidate patches, which can be computationally expensive. Given the large scale of our benchmark and practical resource limitations, we followed the “single-sample” evaluation protocol, as also adopted in prior work such as Multi-SWE-bench [1]. In addition, implementing reranking based on regression testing involves significant infrastructure changes and environment restructuring on the original Agentless code, which falls outside the scope of this work. We will update the results once the conditions are feasible.
>
> We will clarify this limitation in the final version of the paper. We very much look forward to the community or the Agentless author team helping to evaluate their methods on our benchmark.
>
> [1] Zan, Daoguang, et al. "Multi-swe-bench: A multilingual benchmark for issue resolving." arXiv preprint

---

> > ### Comment · Reviewer_cdHm · 2025-08-04
> > **Response to Authors' Rebuttal**
> >
> > Thank you for your detailed rebuttal addressing my concerns. While I appreciate your efforts to respond to each point, I must highlight several areas where the responses remain insufficient.
> >
> > **Positive Acknowledgments:**
> >
> > 1. **Pipeline Reliability**: The quantitative breakdown (379/536 success rate, 12/12 for SWE-bench repos) and failure mode analysis significantly strengthen the paper's technical rigor.
> >
> > 2. **Multi-language Support**: The extension to JavaScript, Rust, Java, and Go is promising, though the timing raises questions about the completeness of the original submission.
> >
> > **Critical Concerns Requiring Attention:**
> >
> > **1. Statistical Robustness - MAJOR CONCERN**
> >
> > Your response essentially argues that computational cost and prior work precedent justify the lack of statistical rigor. This is fundamentally problematic for several reasons:
> >
> > - **Computational cost is not a valid excuse for poor methodology**. Even running a subset (e.g., 100-200 instances) with multiple seeds would provide meaningful confidence intervals.
> > - **The precedent argument is weak**. Leading venues increasingly demand statistical rigor, and "others did it" doesn't justify methodological shortcuts.
> > - **Your benchmark aims to be a community standard**. Setting a low bar for statistical validation undermines this goal.
> >
> > **Recommendation**: At minimum, provide multi-run results on your Lite subset (300 instances) with confidence intervals. This is computationally feasible and scientifically necessary.
> >
> > **2. Evaluation Metrics - PARTIALLY ADDRESSED**
> >
> > Your mention of Patch Apply Rate and Localization Success Rate misses my core concern. These metrics were already in your original paper and still focus on syntactic/structural correctness rather than semantic patch quality. My request for metrics like static code analysis, runtime overhead, or code readability remains unaddressed.
> >
> > **3. Agentless Evaluation - INADEQUATELY ADDRESSED**
> >
> > Your response amounts to "we know the comparison is unfair, but we won't fix it due to implementation challenges." This seriously compromises the benchmark's credibility:
> >
> > - **Incomplete baselines undermine scientific validity**. You cannot claim comprehensive evaluation while knowingly handicapping one method.
> > - **The infrastructure complexity argument is weak**. If implementing proper evaluation is "outside the scope," then perhaps the comparison should be omitted entirely rather than presenting misleading results.
> >
> > **Minor Technical Questions:**
> >
> > 1. **Multi-language timing**: When exactly was multi-language support implemented? If it was available during submission, why wasn't it included in the original paper?
> >
> > 2. **RepoLaunch failure patterns**: Could you provide a more detailed taxonomy of the failure modes beyond the three categories mentioned?
> >
> > **Verdict:**
> >
> > While your work addresses an important problem and introduces valuable automation, the statistical methodology and evaluation completeness issues significantly impact its scientific rigor. For a venue like NeurIPS, these are not minor concerns but fundamental requirements.
> >
> > **Path Forward:**
> >
> > I am willing to maintain my current rating contingent on:
> > 1. **Multi-run statistical analysis** on at least the Lite subset with proper confidence intervals
> > 2. **Complete Agentless evaluation** or explicit removal of incomplete comparisons
> > 3. **Clear limitations section** acknowledging the current methodological constraints
> >
> > Without addressing these core methodological issues, particularly the statistical rigor concern, I would need to reconsider my rating downward.
> >
> > I look forward to seeing these improvements in the final version, as I believe this work has the potential to make a significant impact if executed with appropriate scientific rigor.

---

> > > ### Author Response · Authors · 2025-08-07
> > >
> > > We are very grateful for your comments and detailed suggestions! We have made our best effort to conduct additional experiments, and we would also like to provide several explanations regarding the design in our original work.
> > >
> > > ---
> > >
> > > ### **1. Multi-run experiments**
> > >
> > > Based on your suggestion, we conducted additional multi-run experiments on the Lite subset (300 instances) using SWE-agent + GPT-4.1 with temperature 0.6. The results are as follows:
> > >
> > > | Run 1  | Run 2  | Run 3  | Average | Pass@3  |
> > > |--------|--------|--------|---------|---------|
> > > | 15.33% | 16.67% | 16.67% | 16.22%  | 21.67%  |
> > >
> > > Here, Pass@3 indicates that, if given three attempts, the agent successfully solves 65 out of 300 tasks (21.67%). Compared to the pass@1 score reported in the paper, this represents an improvement of 5.33% in terms of absolute score gains. We will add a dedicated section in the revision discussing these multi-run results and the implications for model evaluation.
> > >
> > > ### **2. Evaluation metrics**
> > >
> > > We appreciate your call for broader metrics beyond those included in the original paper. Since SWE-bench-Live is fundamentally test-driven, running tests directly evaluates semantic correctness, covering some aspects addressed by static analysis. However, we agree that metrics such as runtime overhead and code performance provide valuable complementary insights. We will report per-method runtime profiling in our future benchmark releases and online leaderboard.
> > >
> > > ### **3. Multi-language support**
> > >
> > > At submission, we mainly focused on Python repositories, because Python is objectively the most popular language, and there was also some consideration to follow previous SWE-bench setting which also focus on python only. We believed this work was already very complete, solving long-standing pain points. After submission, we spent additional effort to incrementally adapt RepoLaunch to multiple languages. We agree that this is also very important to scale up SWE tasks and environments in other languages, we will create a multi-language dataset and report models’ scores on other languages in the final revision.
> > >
> > > ### **4. Agentless evaluation**
> > >
> > > Your point regarding baseline completeness is well-taken. We have tried various approaches to reproduce the number of samples used in the official repository (4 localizations and 10 repairs, generating a total of 40 patch samples per task), but found that reranking actually reduced the resolved rate from 12% (single rollout) to 9.67%. The result reported in the paper was based on a single localization and a single repair, which we have clearly stated in the paper. We are considering removing Agentless from the methods we report and instead using fixed localization-repair workflows implemented by other works, such as Kimi-Dev and Agentless-Mini. We remain open to community contributions and would welcome an evaluation from the original authors on SWE-bench-Live, as adapting Agentless poses significantly greater challenges compared to other approaches/agents.
> > >
> > > ### **5. Failure mode analysis and limitations**
> > >
> > > Based on your feedback, we will expand our taxonomy of RepoLaunch failure modes and provide concrete examples and breakdowns in the revision. All logs and experiment trajectories are publicly available to facilitate analysis. We will also strengthen the limitations section to explicitly acknowledge the current constraints and areas for future improvement.
> > >
> > > ---
> > >
> > > Thank you again for your detailed and thoughtful feedback. We are committed to addressing all these points in the final revision and ensuring that SWE-bench-Live sets a rigorous and impactful standard for the community.

---

### Note · Authors · 2025-08-16

We sincerely thank all reviewers for their constructive feedback and positive recognition of SWE-bench-Live’s contributions. Throughout this rebuttal process, we have carefully addressed each concern raised and conducted substantial additional experiments to strengthen our work.

Key improvements made during the rebuttal include: (1) multi-run statistical analysis with Pass@3 metrics on the Lite subset, (2) extension of RepoLaunch to support JavaScript/TypeScript, Rust, Java, and Go, (3) detailed environment setup pipeline reliability statistics with failure mode analysis, (4) creation of a Verified subset with 500 high-quality tasks through LLM-based quality screening, and (5) more latest model results (Qwen3 Coder, Kimi K2, Claude 4 Sonnet). In the final revision, we will integrate all enhancements made during the rebuttal period into the paper, and ensure they are properly consolidated.

We believe SWE-bench-Live addresses critical gaps in existing benchmarks by providing contamination-resistant, continuously updated evaluation that better reflects real-world software development challenges. The dramatic performance drop observed (from 60%+ on static benchmarks to ~20% on our live benchmark) validates the necessity of this work for honest assessment of LLM capabilities.

We are grateful for the reviewers’ engagement in helping us strengthen this work. All code, data, and experimental logs are publicly available, and we look forward to SWE-bench-Live serving as a robust community resource for advancing objective evaluation of LLM coding capabilities.

Thank you for your time and consideration in evaluating our work.

---

### Decision · Program_Chairs · 2025-09-18

**Decision:**

Accept (poster)

**Comment:**

The paper introduces SWE-bench-Live, a live-updatable benchmark for evaluating LLMs and code agents on real-world bug-fixing tasks. The main strengths are the novelty and utility of a continuously updating benchmark, its scalability through automation, and comprehensive evaluations of state-of-the-art models. The weaknesses lie in the initial scope and the need for long-term sustainability of updates, though these do not critically undermine the contribution.

During the rebuttal and discussion, reviewers raised concerns about dataset diversity, reproducibility, and maintenance. The authors provided detailed clarifications and alleviated most concerns. Hence, my final recommendation is ACCEPT.